# Effects of Post-Anthesis Irrigation on the Activity of Starch Synthesis-Related Enzymes and Wheat Grain Quality under Different Nitrogen Conditions

**DOI:** 10.3390/plants12244086

**Published:** 2023-12-06

**Authors:** Lang Xin, Yuanyuan Fu, Shoutian Ma, Caixia Li, Hongbo Wang, Yang Gao, Xingpeng Wang

**Affiliations:** 1College of Water Conservancy and Architecture Engineering, Tarim University, Alar 843300, China; 18942777099@163.com (L.X.); 18083915561@163.com (H.W.); 2Institute of Farmland Irrigation, Chinese Academy of Agriculture Sciences, Xinxiang 453002, China; fyy2016060105@163.com (Y.F.); mashoutian@caas.cn (S.M.); licaixia@caas.cn (C.L.); 3Institute of Western Agricultural, Chinese Academy of Agricultural Sciences, Changji 831100, China

**Keywords:** amylopectin content, interactions between irrigation and nitrogen, soluble sugar content, starch accumulation, winter wheat

## Abstract

To develop optimal management strategies for water and nitrogen fertilizer application in winter wheat cultivation, we conducted a potted experiment to investigate the effects of different irrigation levels and nitrogen fertilizer treatments on the activity of starch synthesis-related enzymes and the grain quality of winter wheat. The potted experiment consisted of three irrigation levels, with the lower limits set at 50–55% (I_0_), 60–65% (I_1_), and 70–75% (I_2_) of the field capacity. In addition, four levels of nitrogen fertilizer were applied, denoted as N_0_ (0 kg N hm^−2^), N_1_ (120 kg N hm^−2^), N_2_ (240 kg N hm^−2^), and N_3_ (300 kg N hm^−2^), respectively. The results revealed the significant impacts of irrigation and nitrogen treatments on the activities of key starch-related enzymes, including adenosine diphosphoglucose pyrophosphrylase (ADPG-PPase), soluble starch synthase (SSS), granule-bound starch synthase (GBSS), and starch branching enzymes (SBE) in wheat grains. These treatments also influenced the starch content, amylopectin content, and, ultimately, wheat yield. In summary, our findings suggest that maintaining irrigation at a lower limit of 60% to 65% of the field capacity and applying nitrogen fertilizer at a rate of 240 kg hm^−2^ is beneficial for achieving both high yield and high quality in winter wheat cultivation.

## 1. Introduction

Winter wheat (*Triticum aestivum* L.) is an essential cereal crop of considerable agronomic and economic importance that plays a central role in food production and security worldwide [1,2]. Efforts to enhance its productivity and resilience are essential to meet the food needs of a growing global population. The grain yield of winter wheat is determined by the interaction between sources and sinks, with source biomass accumulating from photosynthetic products provided by the source organs [3]. The majority of carbohydrates synthesized through plant photosynthesis are transported to the grain in the form of sucrose [4]. Increased sucrose content increases the likelihood of delivering photosynthetic products to the grain, thereby promoting biomass accumulation [5]. When sucrose undergoes decomposition within wheat grains, it serves as a precursor for starch synthesis. Wheat grains are predominantly composed of starch, constituting approximately 65% to 75% of the dry weight of the grains [6]. Starch synthesis in wheat grains is a multifaceted process involving a series of enzyme-catalyzed reactions. Key enzymes such as adenosine diphosphate glucose pyrophosphatase (ADPG-PPase), soluble starch synthase (SSS), granule-bound starch synthase (GBSS), and starch branching enzymes (SBE) play pivotal roles in regulating sucrose metabolism, orchestrating starch synthesis, and governing starch accumulation within wheat grains [7]. Enzymatic activity, which is linked to starch synthesis in the source organs, governs the conversion of sucrose to starch and demonstrates a notable correlation with both the rate of starch accumulation and the dynamics of wheat grain-filling during development [8].

Water and fertilizer are pivotal in facilitating the synthesis, transformation, and transport of photosynthetic products. The combined application of water and nitrogen fertilizer can exert a beneficial and complementary effect on wheat grain yield [9,10]. Insufficient water availability during the grain-filling stage can lead to a reduction in the sucrose content of the plant and will consequently affect the transport of carbohydrates into the wheat grain [11]. Moreover, the activity of key enzymes may be partially inhibited under these circumstances [12]. Several studies have suggested that water deficit can have a positive effect on enhancing the rate of starch accumulation during the early and middle stages of the grain-filling period, leading to the increased activity of enzymes associated with starch synthesis [13,14,15]. Research investigating the influence of nitrogen application on enzymes associated with starch synthesis consistently reveals a pattern: as nitrogen fertilizer application increases, the activity of starch synthesis-related enzymes tends to rise [16]. However, excessive nitrogen application has an adverse effect, causing the wheat to remain green for an extended period and delaying its ripening, which is primarily characterized by a significant reduction in the rate and overall quantity of starch accumulation in wheat grains. This delay is also associated with a decline in the activity of the four enzymes, ultimately resulting in decreased grain weight and yield [17,18].

The flowering and grain-filling periods constitute a pivotal stage in the formation of wheat grain quality, with the supplies of water and nitrogen playing an indispensable role in shaping the development of wheat grain quality [19,20]. Research has shown that the precise timing of irrigation during the flowering and grain-filling stages can significantly enhance wheat grain quality [11]. Adequate water availability during the grain-filling stage promotes larger and heavier wheat grains [21,22]. Insufficient water can lead to smaller and lighter grains, which may reduce the overall grain quality. Water stress during grain-filling can result in a decreased protein content in wheat grains [23]. Reduced protein levels can negatively impact the quality of wheat for baking and other food-processing purposes, as protein is essential for gluten formation and bread quality. Adequate water availability is critical for the accumulation of starch in wheat grains [8]. Water stress can hinder starch synthesis, leading to reduced grain quality in terms of texture, cooking properties, and suitability for processing into various food products [24]. Nitrogen is a primary building block of wheat proteins, particularly gluten, which is critical for baking quality. Adequate nitrogen supply during grain-filling promotes a higher protein content in the grain, contributing to better bread quality [25]. Conversely, insufficient nitrogen can lead to lower protein levels, resulting in reduced dough strength and an inferior bread texture. Nitrogen availability affects the composition and quality of gluten proteins [26]. A proper nitrogen supply leads to the formation of strong and elastic gluten, while inadequate nitrogen can result in weak and extensible gluten [27]. Nitrogen is involved in the synthesis and accumulation of starch in wheat grains, and a moderate reduction in N leads to small increases in the starch content of wheat [28]. Although water and nitrogen application greatly affect the properties of starch and the standard of grain quality, the effects of different N fertilization and irrigation levels on wheat starch quality, combined with yield and grain protein content, remain unclear.

To date, there is no consensus on the influence of water and nitrogen application on starch synthase in wheat grain. How to synergistically improve grain yield and quality through rational irrigation and optimal nitrogen fertilizer application is a pressing issue in the current production of high-quality wheat. Moreover, studies on the interactive effects of water and nitrogen on wheat grain quality remain relatively limited and lack depth, leaving several pertinent questions unanswered. Therefore, the objectives of this study were (1) to investigate the variation in soluble sugar and starch accumulation in winter wheat under different irrigation and nitrogen conditions, (2) to clarify the responses of starch synthase enzyme activity to post-anthesis irrigation under different nitrogen conditions, and (3) to reveal the effects of different water–nitrogen application conditions on wheat grain quality. The results of this study will provide a theoretical basis for the scientific management of irrigation and nitrogen fertilization for winter wheat in the North China Plain and similar areas.

## 2. Results

### 2.1. Effect of Different Water–Nitrogen Treatments on the Soluble Sugar Content of Winter Wheat Grains

During the grain-filling period, the soluble sugar content of winter wheat grains exhibited a noticeable pattern: it decreased rapidly from flowering until the 12th day after flowering, after which it remained at a consistently low level (Table 1).

When considering consistent irrigation conditions, it is worth noting that the wheat grains in N_0_ displayed a significantly lower soluble sugar content compared to the other nitrogen treatments (*p* < 0.05). N_0_ exhibited average decreases in soluble sugar of 13.7%, 23.3%, and 25.2% compared to N_1_, N_2_, and N_3_, respectively. The highest levels of soluble sugar were recorded in N_3_, and there was no statistically significant difference between N_3_ and N_2_. Additionally, under the same nitrogen application conditions, there were significant differences in the soluble sugar between the different irrigation treatments during the initial three sampling days. However, on the last two sampling days, the differences in soluble sugar between the different irrigation treatments were not significant, except for the N_0_ treatment. Overall, it is evident that irrigation, nitrogen fertilizer, and their interaction all exerted substantial influences on the soluble sugar content of the wheat grains in this experiment.

### 2.2. Effect of Different Water–Nitrogen Treatments on Starch Accumulation in Winter Wheat Grains

The starch content in winter wheat grains during the grain-filling period exhibited a consistent increasing trend, peaking between the 24th and 30th days after flowering (Figure 1). Among the various irrigation treatments, the starch content followed the order of I_1_ > I_2_ > I_0_, indicating that reduced irrigation levels constrained starch synthesis, especially when nitrogen fertilizer levels were low. Notably, there was no significant difference in starch content between I_0_N_3_ and I_0_N_2_ (*p* > 0.05); however, both of the treatments yielded significantly higher starch contents than I_0_N_1_ and I_0_N_0_ (*p* < 0.05) between the 18th and 30th days after flowering. Beginning from the 12th day after flowering, differences in the starch content emerged among the different nitrogen treatments, with the order of I_1_N_2_, I_1_N_3,_ I_1_N_1,_ and I_1_N_0_. Specifically, under I_1_ conditions, the starch content in N_2_ was significantly higher than that with N_0_ between the 12th and 30th days after flowering, whereas on the 30th day after flowering, the content in N_3_ surpassed that with the other treatments. Moreover, under I_2_ conditions, the starch content with I_2_N_3_ significantly outperformed the other treatments. To summarize, irrigation, nitrogen fertilizer, and their combined effects all exerted a significant influence on the starch content of wheat grains.

### 2.3. Effect of Different Water–Nitrogen Treatments on Amylopectin Content in Winter Wheat Grains

Amylopectin constituted a substantial portion, ranging from 65–78%, of the total starch content in wheat grains. Its content showed a similar trend to that of total starch accumulation, characterized by a gradual increase followed by a gradual decrease, with the maximum value occurring on the 24th day after flowering (Figure 2). Notably, when subjected to the N_0_ treatment, the amylopectin content in the grains was significantly lower than that of grains subjected to other nitrogen treatments (*p* < 0.05). Starting from the 12th day after flowering, distinctions in the amylopectin content began to emerge among the different treatments, and these differences became particularly significant during the 18th to 24th days after the flowering period. It is worth highlighting that under I_1_N_2_ treatment, the amylopectin content was higher compared to the other nitrogen treatments with equivalent water application. The two-factor ANOVA indicated that irrigation, nitrogen fertilizer, and their combined effects all exerted a significant influence on the amylopectin content of the wheat grain. This suggests that an appropriate balance between water and nitrogen supply plays a crucial role in promoting wheat dry matter accumulation and amylopectin synthesis.

### 2.4. Dynamics of Enzyme Activity in Relation to the Conversion of Sucrose into Starch with Different Water–Nitrogen Treatments

#### 2.4.1. Dynamics of ADPG-PPase Activity under Different Water–Nitrogen Treatments

ADPG serves as a precursor for starch synthesis, with the efficiency of glucose 1-phosphate (G-1-P) conversion to ADPG being directly influenced by the activity of ADPG-PPase. This enzyme activity plays a pivotal role in determining the availability of sufficient precursor molecules for starch synthesis. Figure 3 shows the effect of different water–nitrogen treatments on the activity of ADPG-PPase. Under the I_0_ treatment, the highest activity of ADPG-PPase was recorded on the 18th day after flowering. During the early stages of grain filling, sucrose accumulation took place. However, as the grain filling progressed, ADPG-PPase played a role in the metabolic degradation of sucrose. Notably, the activity of ADPG-PPase was significantly lower with I_0_N_0_ (*p* < 0.05) compared to other treatments, indicating that water and nitrogen stress can substantially reduce ADPG-PPase activity.

In the case of the I_1_ treatment, the ADPG-PPase activity reached its peak on the 24th day after flowering. Furthermore, with the I_1_N_2_ treatment, the ADPG-PPase activity increased by −0.87%, 5.85%, and 24.62% compared to I_1_N_3_, I_1_N_1_, and I_1_N_0_, respectively. This suggested that an appropriate level of water–nitrogen supply could enhance ADPG-PPase activity, thereby facilitating the conversion of wheat grain sucrose to starch. Conversely, excessive nitrogen fertilizer application had a detrimental effect on the ADPG-PPase activity. Notably, there was no significant difference in ADPG-PPase activity between I_2_N_1_ and I_2_N_2_ (*p* > 0.05). However, both treatments exhibited significantly higher activity compared to I_2_N_0_ (*p* < 0.05). The ADPG-PPase activity with I_0_N_2_, I_1_N_2_, and I_2_N_2_ ranged from 444.62 to 684.26 nmol G1P g^−1^ min^−1^, 434.62 to 693.51 nmol G1P g^−1^ min^−1^, and 484.63 to 703.55 nmol G1P g^−1^ min^−1^, highlighting the fact that optimal irrigation conditions maximizes ADPG-PPase activity, whereas inadequate or excessive irrigation levels hinder its increase.

#### 2.4.2. Dynamics of SSS Activity under Different Water–Nitrogen Treatments

SSS, a key enzyme in starch synthesis, exists in a free state in body-building powders and plays a catalytic role in enhancing the reaction between ADPG and starch primers, thereby facilitating the elongation of starch chains. This enzyme is closely associated with the synthesis of amylopectin. SSS activity exhibits a bimodal trend curve during the grain-filling period, reaching its peak on the 12th day after flowering and declining on the 18th day, then rebounding to its maximum value on the 24th day after flowering (Figure 4). Comparing the different treatments, it was observed that the SSS activity was slightly higher in grains treated with I_2_ than in those treated with I_0_ and I_1_. 

When treated with I_0_, the impact of the various nitrogen application levels on SSS activity was notable, with the order of influence being N_2_ > N_3_ > N_1_ > N_0_. Specifically, on the 24th day after flowering, I_0_N_2_ increased the SSS activity by 9.21% compared to I_0_N_3_, by 4.46% compared to I_0_N_1_, and by 15.09% compared to I_0_N_0_, with significant differences being observed between I_0_N_2_ and both I_0_N_3_ and I_0_N_0_. Similarly, for the I_1_N_2_ treatment on the 24th day after flowering, SSS activity increased by 7.9% compared to I_1_N_3_, by 4.5% compared to I_1_N_1_, and by 7.6% compared to I_1_N_0_, again with significant differences. Finally, for the I_2_N_2_ treatment, on the 24th day after flowering, the SSS activity increased by 2.29% compared to I_2_N_3_, by 10.24% compared to I_2_N_1_, and by 18.98% compared to I_2_N_0_, with significant differences being observed compared to I_2_N_1_ and I_2_N_0_. These observations suggest that inadequate or excessive irrigation, as well as suboptimal or excessive nitrogen fertilizer application, have varying impacts on SSS activity. Optimal levels of water and nitrogen supply are critical factors in ensuring the efficient conversion of wheat-grain sucrose to starch.

#### 2.4.3. Dynamics of GBSS Activity under Different Water–Nitrogen Treatments

GBSS performs a role akin to that of SSS and is found in a bound state within body-building powders. This enzyme has the ability to break down glucose molecules from ADPG and transport them to starch precursors, demonstrating a close connection with the synthesis of amylocellulose. Throughout the entire grain-filling period, GBSS activity exhibited an initial increase followed by a subsequent decrease, as illustrated in Figure 5. Notably, the peak of GBSS activity occurred on the 24th day after flowering, aligning with the pattern of amylopectin accumulation.

At a constant level of nitrogen fertilizer application, the GBSS activity followed the order of I_1_ > I_2_ > I_0_ for the three water supply levels. This observation suggests that soil drought not only reduces the capacity of wheat grains to supply photosynthetic products but also impacts GBSS activity, thereby inhibiting the conversion of these products into starch. Moreover, during the grain-filling period, the GBSS activity with I_0_N_2_ surpassed that with I_0_N_0_ and I_0_N_1_. On the 24th day after blossoming, the GBSS activity with I_1_N_2_ increased by 2.26%, 5.59%, and 17.35% compared to that with I_1_N_3_, I_1_N_1_, and I_1_N_0_, respectively. Furthermore, there were significant differences in the GBSS activity between I_1_N_2_ and both I_1_N_3_ and I_1_N_0_ (*p* < 0.05). These findings indicate that an appropriate level of nitrogen fertilizer application can enhance the capacity of wheat-grain photosynthetic products to be converted into starch, thus providing essential support for starch accumulation and product formation.

#### 2.4.4. Dynamics of SBE Activity under Different Water–Nitrogen Treatments

SBE, also known as the Q-enzyme, plays a pivotal role in the synthesis of amylopectin within wheat grains. Its primary function is to create branched sugar chains that work in conjunction with SSS to facilitate the synthesis of amylopectin. As depicted in Figure 6, the SBE activity displayed the characteristic pattern of an initial increase followed by a subsequent decrease, peaking between the 18th and 24th days after flowering. Notably, with the I_0_ treatment, the SBE activity reached its maximum earlier than in the I_1_ treatment, suggesting that soil drought may expedite the grain-filling period.

The SBE activity in the I_0_N_0_ and I_0_N_2_ treatments ranged from 1.18 to 1.76 nmol G1P g^−1^ min^−1^ and from 1.51 to 2.23 nmol G1P g^−1^ min^−1^, respectively. Significant differences in SBE activity were observed between these two treatments (*p* < 0.05). In the I_1_N_2_, I_1_N_0_, and I_2_N_3_ treatments, SBE activity ranged from 1.45 to 2.26 nmol GIP g^−1^ min^−1^, 1.22 to 1.86 nmol GIP g^−1^ min^−1^, and 1.52 to 2.03 nmol G1P g^−1^ min^−1^, respectively. These results indicated that excessive water and nitrogen supply can impede the increase in SBE activity, whereas an optimal water–nitrogen supply condition can enhance SBE activity and facilitate starch synthesis.

### 2.5. Effect of Different Water–Nitrogen Treatments on the Grain Quality and Yield of Winter Wheat

Table 2 reveals that the water treatments exhibited a noteworthy effect on thousand-grain weight and overall yield. In contrast, the nitrogen fertilizer demonstrated a significant or even extremely significant influence on the thousand-grain weight and yield. Notably, the interaction between water and nitrogen exerted a significant influence on wheat yield.

Across the various nitrogen fertilizer treatments, the order of impact on the thousand-grain weight and yield was as follows: N_2_ > N_3_ > N_1_ > N_0_. This suggests that increasing the application of nitrogen fertilizer can increase wheat yield, while excessive or insufficient nitrogen fertilizer application can hinder yield improvement. Specifically, with the I_1_N_2_ treatment, wheat yield increased by 46.97% compared to I_0_N_0_, by 7.56% compared to I_1_N_3_, and by 6.65% compared to the I_2_N_3_ treatments. While the highest grain yield was observed with the I_2_N_2_ treatment, it did not significantly differ from that with the I_1_N_2_ treatment. This indicates that increased irrigation and nitrogen fertilizer application can indeed boost grain yield, but the excessive use of water and nitrogen fertilizer does not yield significant improvements and instead leads to the wastage of these resources.

Irrigation and nitrogen fertilizer had significant impacts on the protein content in the wheat grains, while their interaction did not notably affect the protein content (Table 2). The protein content increased with higher irrigation levels and nitrogen fertilizer amount, except in the case of I_1_N_3_. The highest value, recorded with I_1_N_2_, was 20% higher than that with I_1_N_3_. The irrigation treatments exhibited significant effects on the wet gluten content (*p* < 0.01), with higher levels of irrigation in the water resulting in an increased wet gluten content (Table 2). Generally, an increase in nitrogen application led to a higher wet gluten content, while the nitrogen treatments did not have a significant impact on the wet gluten content (*p* > 0.05). Irrigation had a significant effect on the sedimentation value in wheat grains (*p* < 0.01), while nitrogen fertilizer application and the interaction between irrigation and the nitrogen fertilizer did not have a significant impact on the sedimentation value (Table 2). The sedimentation value was increased with the increase in irrigation and nitrogen applications. The values for I_1_N_2_, I_1_N_3_, I_2_N_2_, and I_2_N_3_ were significantly greater than those for the other treatments, while the differences between the four treatments were not statistically significant. 

## 3. Discussion

After flowering, the sucrose that is synthesized in the leaves of wheat is transported via the phloem to the grain, where sucrose metabolism is mainly a catabolic reaction, i.e., the breakdown of sucrose into the raw material for starch synthesis, catalyzed by a number of enzymes. Water and nitrogen status have significant impacts on soluble sugar and starch, starch synthesis-related enzyme activity, and the grain yield and quality of winter wheat. 

### 3.1. The Effects of Different Water and Nitrogen Treatments on the Activity of Starch Synthesis-Related Enzymes

Previous studies have shown that the supply capacity of plant sources generally meets the grain demand, while the storage capacity of the grain and the ability to transport substances may be the major factors limiting the accumulation of dry matter [29,30]. In terms of sink activity, the results of this study confirmed that ADPG-PPase, SSS, GBSS, and SBE are key enzymes that regulate the metabolism of soluble sugars in wheat grains and the synthesis and accumulation of starch. Ahmadi and Baker [31], as well as Thitisaksakul et al. [32], suggested that controlled water stress could decrease the activity of ADPG-PPase, GBSS, and SSS and that the activity of ADPG-PPase and SSS could significantly influence the progress of grain filling. The results of this experiment also demonstrated that water stress decreased the activity of these three enzymes and expedited the time of their peak appearance. However, some studies have suggested that soil drought may increase the rate of starch accumulation and the activity of related enzymes in the early stages of grain filling [33]. It was believed that the reduction in enzyme activity during the later stages was primarily due to the beneficial impacts of drought in the early stages, which facilitated the transport of non-structural carbohydrates into the grain. Elevating nitrogen levels was found to be beneficial for increasing the activity of ADPG-PPase, SSS, GBSS, and SBE; however, the degree of increased activity varied for each enzyme [34]. Among these enzymes, GBSS and SBE exhibited the greatest increase in enzymatic activity, with activities under treatment I_1_N_2_ increasing by 11.9% to 28.9% and 10.5% to 15.5%, respectively, compared to that with I_1_N_3_ and I_1_N_0_. The above results align with the findings of Yue et al. [35], who suggest that excessive nitrogen fertilizer tends to decrease enzymatic activity, while appropriately increasing the nitrogen levels aids in enhancing the activity of the enzymes related to starch synthesis [34]. 

### 3.2. The Effects of Different Water and Nitrogen Treatments on the Grains’ Soluble Sugar and Starch Content

After the flowering stage of wheat, the sucrose that is synthesized in the leaves is transported through the phloem to the developing grains. Within the grain, sucrose metabolism predominantly involves degradation reactions. Through a series of enzymatic processes, sucrose was broken down into components that serve as sources for starch synthesis [36]. In this experimental context, we observed a consistent decrease in the soluble sugar content within the grains as the grain-filling process progressed. This phenomenon indicated that a direct relationship existed between starch content and soluble sugar content, with the increase in starch content being a direct consequence of sucrose conversion [37]. As the nitrogen levels increased, there was a corresponding rise in both soluble sugar and starch contents. This finding aligned with the conclusions reached by Yu et al. [38] and Zhang et al. [39], who observed that elevating the nitrogen levels facilitated the transport of stored dry matter from the nutritional organs to the grain. This increase in nitrogen levels had a positive impact on the sucrose content of the grain by increasing the availability of substrates for starch synthesis. Nevertheless, some viewpoints suggest that excessive nitrogen levels may have the opposite effect, leading to a reduction in both sucrose and starch contents within the grain [40]. This phenomenon primarily arose from the disruption of the delicate balance between carbon and nitrogen within individual plants. Elevated nitrogen levels tended to divert more carbon resources toward nitrogen metabolic processes [41]. Consequently, when synthesizing an equivalent volume of proteins, a substantial portion of the starch reserves can be consumed. As a result, high nitrogen treatments often result in increased grain protein content and decreased starch content [42]. Moreover, excessive nitrogen can also inhibit the photosynthetic rate within grain blades, primarily due to non-stomatal factors, thereby influencing the synthesis and transport of photosynthetic products [43]. Abid et al. [13] discovered that subjecting wheat to a specific level of drought stress during the grain-filling stage could expedite the conversion of sucrose into starch. This not only shortened the grain-filling period but also promoted earlier maturation. The results of this study further indicated that water stress had the effect of hastening the appearance of the soluble sugar peak, expediting the conversion of sucrose to starch, and enhancing the overall transport rate.

### 3.3. The Effects of Different Water and Nitrogen Treatments on Grain Quality and the Yield of Winter Wheat

The process of wheat grain-filling is a vital physiological stage that is closely linked to grain yield. Among the various factors influencing the growth and development of wheat, water and nitrogen play pivotal roles. Previous studies have indicated that soil water deficiency can diminish the conversion capacity of photosynthetic products in winter wheat during the grain-filling period, leading to a weakened grain sink [44]. Conversely, the judicious application of nitrogen fertilizer can facilitate the accumulation and transport of dry matter in winter wheat, ultimately resulting in a high yield [45]. The experimental results revealed that water had a minimal impact on both the wheat panicle number and grain number per panicle. However, under moderate and high-water treatments, the addition of a nitrogen fertilizer led to a significant increase in both wheat panicle number and grain number per panicle as well as the thousand-grain weight (Table 2). Furthermore, this nitrogen fertilizer application substantially boosted wheat yield, which aligned closely with the results reported by Tan et al. [46]. 

Previous studies have indicated that increasing the volume of irrigation can promote the accumulation of grain proteins [47], while other research has suggested that drought conditions are not conducive to increasing the grain’s protein content [48]. The results of this study demonstrated that appropriate irrigation management had a positive impact on increasing wheat grain protein content. Additionally, the application of nitrogen fertilizer significantly enhances the contents of grain proteins and their various components [35]. However, it should be noted that the main reason for decreased quality when increasing irrigation frequency was the reduction in insoluble glutenin, with the impact on quality traits being particularly pronounced after flowering [47]. The above results indicate that under the interaction of water and nitrogen conditions, changes in wheat grain protein content are associated with various protein components, particularly changes in soluble proteins and gliadins, which play a crucial role in determining protein processing quality. Therefore, in high-yielding and high-quality crops, under appropriate nitrogen application, the regulation of both irrigation amount and frequency can be employed to control both grain yield and protein composition, which can help to achieve higher grain yields and superior grain quality.

The results of this study indicated a significant correlation between the highest levels of enzymatic activity of ADPG-PPase, SSS, SBE, and GBSS and the thousand-grain weight (*r* > 0.95 **). Adequate water and nitrogen fertilizer supply during the grain-filling stage substantially increased the activity of starch synthesis-related enzymes in the wheat grains. This, in turn, amplified the utilization of photosynthetic products for grain development, facilitated the conversion of grain sucrose into starch, promoted grain-filling, and ultimately led to an increased grain weight. Furthermore, the application of nitrogen fertilizer resulted in a substantial boost in overall yield. However, an inadequate water supply and excessive nitrogen levels had the opposite effect, inhibiting the activity of the ADPG-PPase, SSS, SBE, and GBSS enzymes, consequently reducing grain starch accumulation and diminishing wheat yield. To optimize wheat yield during the grain-filling phase, it was crucial to maintain soil irrigation at a lower limit of 60% to 65% of the field’s water capacity and apply nitrogen fertilizer at a rate of 240 kg hm^−2^, which was confirmed by field experiments with winter wheat [10,49]. This combination increased the sucrose content of the grain, ensured a sufficient supply of source organs, and enhanced starch synthetase activity in the grains. Ultimately, it improved the carbon and nitrogen metabolism in wheat, resulting in higher wheat yields.

## 4. Materials and Methods

### 4.1. Plant Materials and Site Description

The winter wheat (*Triticum aestivum* L.) variety of Zhoumai-22 was cultivated in pots under a rainproof shelter that was located in the Qiliying Experimental Station of the Chinese Academy of Agricultural Sciences in Xinxiang City (35°08′ N, 113°45′ E), Xinxiang City, China. The experiment was conducted in the 2022–2023 winter wheat season; the climate factors during the wheat season are shown in Figure 7. Each pot had a diameter of 40 cm and a height of 60 cm. Loam soil, which is the typical soil of the region, was used to fill the pots; it had a bulk density of 1.45 g cm^−3^ and a pH of 8.4. The contents of available nitrogen, phosphorus, and potassium fertilizer were 36.73, 21.88, and 105.48 mg kg^−1^, respectively. The soil had a field capacity of 0.30 m^3^ m^−3^, which was estimated from the water characteristic curve.

### 4.2. Experimental Design

The experiment was conducted in a split-plot design, with the primary plot being subjected to nitrogen fertilizer application and the secondary plot receiving water application. The nitrogen fertilizer was applied at four levels: N_0_ (0 kg hm^−2^), N_1_ (120 kg hm^−2^), N_2_ (240 kg hm^−2^), and N_3_ (300 kg hm^−2^). The ratio of basal nitrogen to top-dressed nitrogen was 50:50. The basal nitrogen fertilizer was applied before the wheat was sown, and the top-dressed nitrogen was applied at the early jointing stage. The irrigation and nitrogen levels were designed according to the previous results [50,51]. The three levels of irrigation were set as follows: I_0_ (irrigation lower limit of 50~55% of the field capacity), I_1_ (irrigation lower limit of 60~65% of the field capacity), and I_2_ (irrigation lower limit of 70~75% of the field capacity). A full combination of the application levels of water and nitrogen led to 12 treatments, and each treatment was replicated in triplicate. 

The winter wheat was planted on October 18, with a row spacing of 10 cm and a seeding depth of 3–5 cm. All the treatments in the period from seeding to flowering were supplied along with sufficient water (the irrigation lower limit was 70–75% of the field capacity, and the irrigation rate was 36 mm); after flowering, the irrigation treatments were carried out according to the study design. Before flowering, water holding was carried out to control the soil moisture under the different treatments according to the design value. After flowering (20 April), irrigation was applied on the 8th, 16th, and 26th days after flowering at an irrigation rate of 36 mm. Phosphorus and potassium fertilizers for each treatment were applied all at once, at a dose of 120 kg hm^−2^ for P_2_O_5_ and 105 kg hm^−2^ for K_2_O—a fertilizer application level that was based on the results of soil testing and formulated fertilization. 

When the winter wheat was flowering, wheat panicles with same-day blossoms and a nearly similar growth status were selected and marked, and grains in the middle segment of the wheat panicles were selected on the 6th, 12th, 18th, 24th, and 30th days after flowering, respectively. One portion of these grains was placed in liquid nitrogen and immediately frozen at −80 °C in a refrigerator, for the measurement of enzymatic activity, while the other portions of these grains were placed in an oven at 105 °C for 30 min and were then baked at 70 °C until they had a constant weight, followed by measurement of the sucrose and starch contents of the grains, which was replicated in triplicate.

### 4.3. Measurements

#### 4.3.1. Contents of Soluble Sugar and Starch in Wheat Grains

After the grain biomass was detected, the grains were crushed through a 100-mesh sieve. The soluble sugar and starch contents were measured using anthrone colorimetry following a previously described method [52]. Briefly, 0.2 g of dried grain samples were placed into a 15 mL tube, and 6 mL of 80% ethanol was added. The samples were extracted in a water bath at 80 °C for 30 min, then centrifuged (3000× *g*) for 5 min, and the supernatant was collected. The supernatant was collected three times and transferred to a beaker to measure the total volume of the extracted solution in the sample. The ethanol was evaporated to 2–3 mL in a constant-temperature water bath at 85 °C and was then transferred to a 50 mL volumetric flask with distilled water. One milliliter of the supernatant and 5 mL of anthrone reagent were added to the mixture. The mixture was then decanted and cooled in a water bath for 10 min. The optical density (OD) was measured at 625 nm and the sugar content of the extract was determined from the standard curve [52,53]. The residue was moved into a 50 mL flask with 20 mL of distilled water, and this mixture was placed in a boiling water bath for 15 min. After cooling, 2 mL of cold 9.2 mol/L perchloric acid was added, then the mixture was stirred continuously, and distilled water was added to a final volume of 10 mL, mixed, and centrifuged for 10 min. The supernatant was poured into a 50 mL volumetric flask. Then, 2 mL of 4.6 mol/L perchloric acid was added to the precipitate and stirred for 15 min. Subsequently, water was added up to 10 mL and the mixture was mixed and centrifuged (5000 g) for 10 min, then the supernatant was collected and transferred into a 50 mL volumetric flask with distilled water to quantify the starch content. At a wavelength of 625 nm, the OD was measured via blank zero adjustment, and the starch content was determined from the standard curve [52]. The total soluble sugar and starch contents (the non-structural carbohydrates (NSCs)), which represent the sum of these two substances, and their ratio were also calculated.

#### 4.3.2. Enzymatic Activity in Wheat Grains

For the extraction of crude enzymes and the plotting of standard curves, 10 grains in the middle segment of the wheat panicles were selected and weighed. They were then ground into powder under liquid nitrogen, followed by the addition of 5 mL of extraction solution to the powder. This was then ground down into a homogenous slurry, which was centrifuged at a speed of 10,000 r min^−1^ for 10 min. The supernatant was collected and bathed in ice and was then used as a crude enzyme-extraction solution for the measurement of the activity of ADPG-PPase, SSS, and SBE. A 5-mL extraction solution was added to the precipitate and shaken until a homogenous mixture was achieved, which was fully suspended, in order to measure GBSS activity. Standard solutions of glucose-1-phosphate (G-1-P) with a series of concentrations of 0, 100, 200, 400, 600, 800, and 1000 nmol ml^−1^ were prepared, and a standard curve was generated. 

Enzymatic activity measurement: ADPG-PPase activity measurement was based on the method described by Chen et al. [54]. A 20-μL crude enzyme extraction solution was added to a 110-μL reaction solution, followed by a reaction at 30 °C for 20 min, then the resulting mixture was placed in freshly boiled water for 30 s to terminate the reaction, followed by 10 min of centrifugation at a speed of 10,000 r min^−1^. The supernatant of 100 μL was selected and added to a 5.2-μL colorimetric solution (5.76 mmol L^−1^ NADP (icotinamide adenine dinucleotide phosphate), 0.08 unit of P-glucomutase (phosphogluco-mutase), and 0.07 unit of G6P-dehydrogenase (glucose-6-phosphate dehydrogenase), followed by a reaction at 30 °C for 10 min, after which the absorbance at 340 nm was measured. In the meantime, the standard solutions were measured. 

The measurements of SSS, GBSS, and SBE activity were based on the method used by Nakamura and Yuki [55]. Enzymatic activity was expressed via the amount of G-1-P produced per unit weight of samples in a 1-minute reaction. The activity was measured in triplicate to generate the mean values for use. The extraction solution was a mixture of 100 mmol L^−1^ of Tricine-NaOH (pH = 8.0), 8 mmol L^−1^ of MgCl_2_, 2 mmol L^−1^ of EDTA, 12.5% (*v*/*v*) glycerol, and 50 mmol L^−1^ of 2-Mercaptoethanol. The reaction solution was a mixture of 100 mmol L^−1^ of Hepes-NaOH (pH = 7.4), 1.2 mmol L^−1^ of ADPG, 3 mmol L^−1^ of PPi, 5 mmol L^−1^ of MgCl_2_, and 4 mmol L^−1^ of DTT.

#### 4.3.3. Measurement of Grain Quality and Yield

After the wheat ripened, a 1 m^2^ quadrant was selected and the panicle number, grain number per panicle, thousand-grain weight, and grain yield were measured, respectively. The measurement was repeated three times, and the mean value was used. 

The total protein content (dry basis), wet gluten content, fatty acid content, and sedimentation value in the wheat grain were determined by the Institute of Quality Standards, Henan Academy of Agricultural Sciences. Total protein content was measured using the GB 5009.5-2016 First Method and was calculated by multiplying a conversion factor of 5.7 with the percentage of N in grain. Wet gluten content was determined via GB/T 5506.2-2008. It was expressed on a 14% moisture basis. Each parameter was determined with three biological replicates.

### 4.4. Data Analysis

SPSS 20.0 (IBM Corp., Armonk, NY, USA) was used for the data processing and analysis of variance (ANOVA), and the significant differences were determined using Duncan’s method at the level of *p* < 0.05. The two-way ANOVA was applied to test irrigation treatment (I), nitrogen treatment (N), and the interaction of I × N.

## 5. Conclusions

The grain-filling stage is a critical period for the development of wheat yield and grain quality. The present results indicated that post-anthesis irrigation and N fertilizer status during the grain-filling stage significantly affected wheat starch accumulation, starch-related enzyme activities, and the yield and quality. The results suggest that irrigation management at the lower limit of 65% of the field capacity, combined with 240 kg hm^−2^ of N fertilizer could contribute to the improvement of wheat yield and quality. The effect of irrigation management on the carbon and nitrogen metabolism of wheat kernels should be investigated in the context of the split application of nitrogen fertilizer in the future.

## Figures and Tables

**Figure 1 plants-12-04086-f001:**
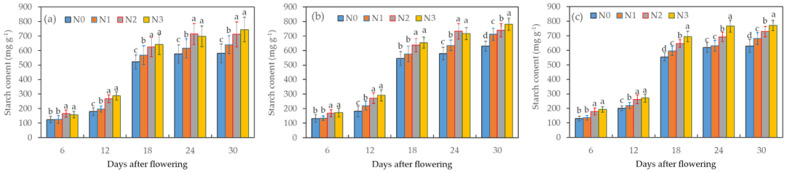
Effects of different treatments of irrigation (**a**—I_0_, **b**—I_1_, and **c**—I_2_) and nitrogen fertilizer on the starch content in wheat grains during the grain-filling period. Data are the mean ± standard deviation (n = 3). The different letters on top of the error bars represent significant differences (*p* < 0.05).

**Figure 2 plants-12-04086-f002:**
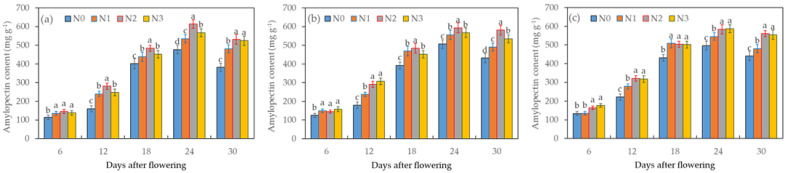
Effects of the different treatments of irrigation (**a**—I_0_, **b**—I_1_, and **c**—I_2_) and nitrogen fertilizer on the amylopectin content in wheat grains during the grain-filling period. Data are the mean ± standard deviation (n = 3). The different letters on top of the error bars represent significant differences (*p* < 0.05).

**Figure 3 plants-12-04086-f003:**
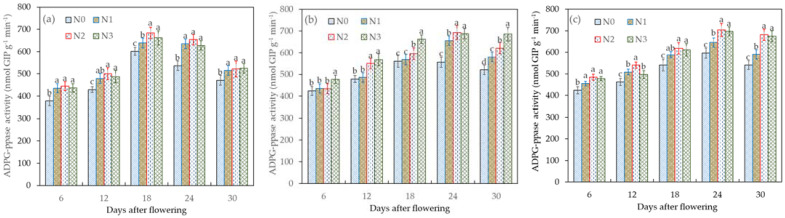
Effects of the different treatments (**a**—I_0_, **b**—I_1_, and **c**—I_2_) on ADPG-PPase activity in wheat grains during the grain-filling period. Data are the mean ± standard deviation (n = 3). The different letters on top of the error bars represent significant differences (*p* < 0.05).

**Figure 4 plants-12-04086-f004:**
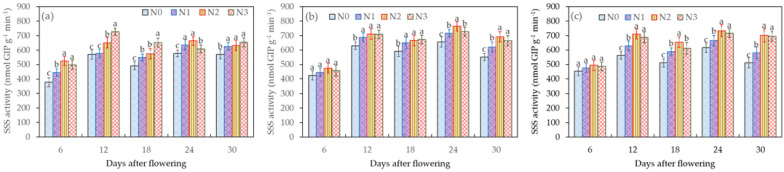
Effects of the different treatments (**a**—I_0_, **b**—I_1_, and **c**—I_2_) on SSS activity in wheat grains during the grain-filling period. Data are the mean ± standard deviation (n = 3). The different letters on top of the error bars represent significant differences (*p* < 0.05).

**Figure 5 plants-12-04086-f005:**
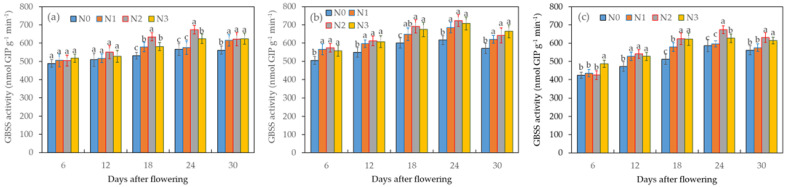
Effects of the different treatments (**a**—I_0_, **b**—I_1_, **c**—I_2_) on GBSS activity in wheat grains during the grain-filling period. Data are the mean ± standard deviation (n = 3). The different letters on top of the error bars represent significant differences (*p* < 0.05).

**Figure 6 plants-12-04086-f006:**
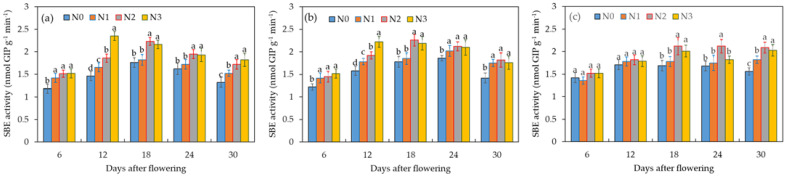
Effects of different treatments (**a**—I_0_, **b**—I_1_, **c**—I_2_) on SBE activity in wheat grains during the grain-filling period. Data are the mean ± standard deviation (n = 3). The different letters on top of the error bars represent significant differences (*p* < 0.05).

**Figure 7 plants-12-04086-f007:**
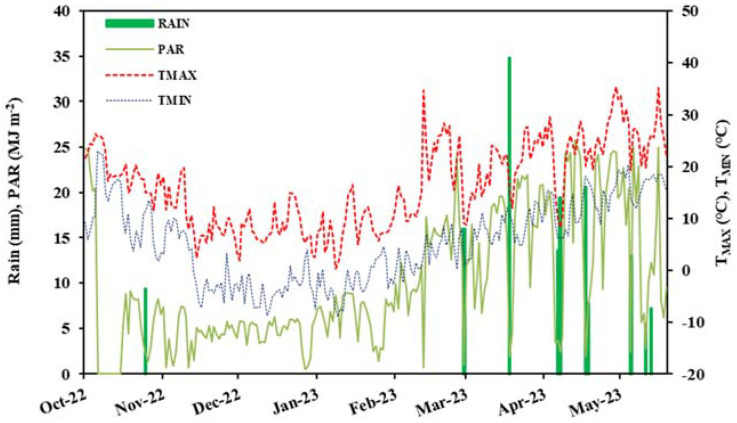
Rain, PAR, T_max_, and T_min_ during the winter wheat growing season of 2022–2023.

**Table 1 plants-12-04086-t001:** The soluble sugar content (mg g^−1^ FW) in wheat grains during the grain-filling period under different treatments of nitrogen fertilizer and irrigation.

Treatment	Days after Flowering				
	6	12	18	24	30
I_0_N_0_	93.57 ± 6.28 c	65.94 ± 5.84 c	51.25 ± 3.54 c	38.52 ± 2.56 b	30.21 ± 2.74 d
I_0_N_1_	99.48 ± 8.73 b	78.61 ± 6.49 b	64.18 ± 4.51 b	46.25 ± 4.84 a	39.51 ± 2.68 c
I_0_N_2_	117.82 ± 8.41 a	90.21 ± 6.54 a	81.41 ± 5.24 a	47.82 ± 5.28 a	38.52 ± 2.41 b
I_0_N_3_	115.23 ± 6.75 a	92.14 ± 7.51 a	85.42 ± 8.77 a	51.21 ± 4.86 a	41.25 ± 3.41 a
I_1_N_0_	102.24 ± 6.54 c	84.63 ± 6.54 c	56.21 ± 4.91 c	43.56 ± 3.61 b	36.34 ± 2.89 c
I_1_N_1_	108.24 ± 9.77 b	94.26 ± 7.48 b	68.62 ± 6.21 b	48.26 ± 4.52 ab	41.61 ± 4.91 b
I_1_N_2_	128.34 ± 11.44 a	111.54 ± 8.54 a	83.32 ± 5.21 a	52.61 ± 5.61 a	42.14 ± 4.11 a
I_1_N_3_	125.11 ± 14.51 a	110.25 ± 8.75 a	82.31 ± 7.84 a	55.32 ± 5.14 a	48.26 ± 3.62 a
I_2_N_0_	145.21 ± 13.55 c	124.52 ± 12.28 c	61.25 ± 4.21 c	45.26 ± 5.22 b	35.21 ± 2.11 b
I_2_N_1_	160.82 ± 16.24 b	134.21 ± 11.29 b	88.62 ± 6.24 ab	47.65 ± 3.65 b	41.26 ± 3.15 a
I_2_N_2_	190.26 ± 15.62 a	142.61 ± 14.63 a	94.26 ± 7.22 a	57.41 ± 3.28 a	45.62 ± 4.65 a
I_2_N_3_	188.24 ± 17.21 a	145.26 ± 11.34 a	93.84 ± 4.21 a	56.22 ± 4.58 a	48.93 ± 3.81 a
I	**	**	**	**	*
N	**	**	**	**	**
I × N	**	**	**	**	**

Note: Data are the mean ± standard deviation (n = 3). Different alphabets on each sampling date under the same irrigation treatment represent significant differences (*p* < 0.05). *: *p* < 0.05, **: *p* < 0.01.

**Table 2 plants-12-04086-t002:** Effects of different irrigation and nitrogen fertilizer treatments on yield and the yield components of winter wheat.

Treatment	Protein Content (g 100 g^−1^)	Sedimentation Value (mL)	Wet Gluten Content (%)	Thousand-Grain Weight (g)	Yield (kg hm^−2^)
I_0_N_0_	13.24 e	30.56 f	32.21 e	40.42 f	6298.22 g
I_0_N_1_	15.23 d	32.52 e	34.67 d	40.34 e	7208.23 e
I_0_N_2_	17.62 c	34.25 d	36.36 c	43.11 c	7711.42 d
I_0_N_3_	20.21 b	34.21 d	38.10 ab	43.25 c	7506.48 de
I_1_N_0_	14.36 e	36.25 c	34.31 d	40.91 d	6322.01 fg
I_1_N_1_	16.38 c	42.26 b	37.21 c	44.85 bc	8080.11 c
I_1_N_2_	23.21 a	46.26 a	42.52 a	49.77 a	9256.72 a
I_1_N_3_	19.32 b	47.21 a	41.45 a	45.24 b	8605.79 b
I_2_N_0_	16.25 c	36.61 c	34.51 d	40.02 d	6275.12 f
I_2_N_1_	17.26 c	40.12 b	36.77 c	44.83 bc	8136.15 c
I_2_N_2_	19.26 b	45.26 a	38.52 ab	48.09 a	9041.50 a
I_2_N_3_	20.36 b	48.52 a	41.25 a	46.25 b	8679.61 b
I	0.000 **	0.000 **	0.000 **	0.024 *	0.031 *
N	0.038 *	0.079	0.062	0.001 **	0.000 **
I × N	0.058	0.101	0.092	0.027 *	0.006 **

Different letters in the same column indicate significant differences among treatments at the 0.05 level. * *p* < 0.05; ** *p* < 0.01.

## Data Availability

The dataset will be available on request.

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
