# Peer review of "Effects of Post-Anthesis Irrigation on the Activity of Starch Synthesis-Related Enzymes and Wheat Grain Quality under Different Nitrogen Conditions"

_plants, 2023, doi:10.3390/plants12244086_

Round 1

Reviewer 1 Report

Comments and Suggestions for Authors

The manuscript entitled "Effects of post-anthesis supplementary irrigation on the activity of starch synthesis-related enzymes and grain quality of winter wheat under different nitrogen conditions”  is relevant to the Plants journal (section Crop Physiology and Crop Production/Special Issue: Strategies to Improve Water-Use Efficiency in Plant Production) but needs improvements before publication in this journal.

Generally, the paper is relatively straightforward and well-written. I think, the Authors could consider the shorter title of the manuscript.  The Authors should improve Keywords and remove some title repetitions. The Authors could also more underline the purpose of the study. The Authors should improve Material and methods section  and provide more details on plant material (wheat cultivar, add also latin name), growth conditions (e.g. temperature, light) and some estimation conditions (e.g. sugars – add extraction procedure). Presentation and description of some results could be more carefully checked in  Results section (e.g., all figs. “c” - line X description). I think, the Authors could modify the beginning of Discussion. Also the References list and citations must be more carefully checked. In addition, there are small mistakes in the text of manuscript (e.g., lines:  24, 46, 90, 93, 104, 184, 315, 457, 458, 492, 609, 613), that need to be corrected by Authors.

Author Response

The manuscript entitled "Effects of post-anthesis supplementary irrigation on the activity of starch synthesis-related enzymes and grain quality of winter wheat under different nitrogen conditions” is relevant to the Plants journal (section Crop Physiology and Crop Production/Special Issue: Strategies to Improve Water-Use Efficiency in Plant Production) but needs improvements before publication in this journal.

Generally, the paper is relatively straightforward and well-written. I think, the Authors could consider the shorter title of the manuscript.  The Authors should improve Keywords and remove some title repetitions. The Authors could also more underline the purpose of the study. The Authors should improve Material and methods section and provide more details on plant material (wheat cultivar, add also latin name), growth conditions (e.g. temperature, light) and some estimation conditions (e.g. sugars – add extraction procedure). Presentation and description of some results could be more carefully checked in Results section (e.g., all figs. “c” - line X description). I think, the Authors could modify the beginning of Discussion. Also the References list and citations must be more carefully checked. In addition, there are small mistakes in the text of manuscript (e.g., lines:  24, 46, 90, 93, 104, 184, 315, 457, 458, 492, 609, 613), that need to be corrected by Authors.

Response: Thanks for your comments and suggestions. (1) The title has been revised to read ‘Effects of post-anthesis irrigation on the activity of starch syn-thesis-related enzymes and wheat grain quality under different nitrogen conditions’ (Lines 2-4). (2) Keywords were revised as ‘Amylopectin content; Interactions between irrigation and nitrogen; Soluble sugar content; Starch accumulation; Winter wheat’ (Lines 24-25). (3) The purpose of this study was revised according to your suggestions (Lines 90-102). (4) Wheat cultivar of Zhoumai-22 and Latin name were added (Line 407). Winter wheat growth conditions (temperature and radiation) were shown in Fig.7. Procedure for soluble sugar and starch measurements was provided (Lines 453-472), and the procedure for measuring enzymatic activity was checked. All figures were checked and revised. (5) The Discussion was revised according to your comments (Lines 297-302). (6) References were checked. (7) We checked the MS carefully and eliminated grammatical and spelling errors.

Reviewer 2 Report

Comments and Suggestions for Authors

Review of “Effects of post-anthesis supplementary irrigation on the activity of starch synthesis-related enzymes and grain quality of winter wheat under different nitrogen conditions”

 In this manuscript, the authors reported their pot experiment on the effects of different irrigation levels and nitrogen fertilizer treatments on the activity of starch synthesis-related enzymes and the grain quality of winter wheat. They tested three different levels of irrigation and four levels of nitrogen fertilizer and their effects on the activities of a few starch-related enzymes. From statistical analysis, they concluded one of the 12 combinations of irrigation and fertilizer as the most optimal. My main comments are as follows.

1. Although there was no Conclusion section, the main conclusion on the optimal level of irrigation and fertilizer appeared to only confirm the results from previous studies as stated in the Discussion section. In addition, this conclusion did not seem to be very strong as compared to other combinations from reading the manuscript.

2. The organization of the manuscript was not optimal to read. While the structure by placing the results before methods might be the preferred format by the journal (?), I found this made the reading interruptive. The readers would have to move back and forth between the results and materials and methods or would have to read section 4 first before sections 2 and 3. In addition, no separate section of conclusions made the comprehension of big-picture contributions difficult.

3. The section of Materials and Methods was not well articulated. There were no justifications on why the stated levels of irrigation and fertilizer. To be convincing, I think the authors should discuss why these levels were used. The pot experiment itself was not well described either. The pot has a diameter of 40 cm and wheat was planted with a row spacing of 20 cm. There were a few wheats in one pot or the spacing between pots was 20 cm? While the levels of irrigation and fertilizer were specified, no scheduling was given. Before the flowering, sufficient water was supplied. After flowering, what was the schedule?

4. The irrigation was set at certain percentage of field capacity. How was the field capacity of soil determined? Why loam soil only? Fertilizer application level was based on the results of soil testing. What testing?

5. In the Abstract, all fertilizer level for all treatments were mistaken as 0 kg N hm-2.

6. The language was generally fine, but there were occasional grammatical errors. Some minor editing is required by the authors.

In summary, I recommended major revision. 

Comments on the Quality of English Language

The language was generally fine, but there were occasional grammatical errors. Some minor editing is required by the authors.

Author Response

Review of “Effects of post-anthesis supplementary irrigation on the activity of starch synthesis-related enzymes and grain quality of winter wheat under different nitrogen conditions”

 In this manuscript, the authors reported their pot experiment on the effects of different irrigation levels and nitrogen fertilizer treatments on the activity of starch synthesis-related enzymes and the grain quality of winter wheat. They tested three different levels of irrigation and four levels of nitrogen fertilizer and their effects on the activities of a few starch-related enzymes. From statistical analysis, they concluded one of the 12 combinations of irrigation and fertilizer as the most optimal. My main comments are as follows.

  1. Although there was no Conclusion section, the main conclusion on the optimal level of irrigation and fertilizer appeared to only confirm the results from previous studies as stated in the Discussion section. In addition, this conclusion did not seem to be very strong as compared to other combinations from reading the manuscript.

Response: Thanks for your comments. The conclusion section was added to the MS (Lines 521-529).

  1. The organization of the manuscript was not optimal to read. While the structure by placing the results before methods might be the preferred format by the journal (?), I found this made the reading interruptive. The readers would have to move back and forth between the results and materials and methods or would have to read section 4 first before sections 2 and 3. In addition, no separate section of conclusions made the comprehension of big-picture contributions difficult.

Response: The structure is provided by the template of the journal. According to your suggestion, the conclusion section was added to the MS (Lines 521-529).

  1. The section of Materials and Methods was not well articulated. There were no justifications on why the stated levels of irrigation and fertilizer. To be convincing, I think the authors should discuss why these levels were used. The pot experiment itself was not well described either. The pot has a diameter of 40 cm and wheat was planted with a row spacing of 20 cm. There were a few wheats in one pot or the spacing between pots was 20 cm? While the levels of irrigation and fertilizer were specified, no scheduling was given. Before the flowering, sufficient water was supplied. After flowering, what was the schedule?

Response: (1) Irrigation and nitrogen fertilizer levels were designed according to the previous results (Lines 425-426), and the references were added (Lines 666-670). (2) It was a mistake for wheat row spacing, it should be 10 cm (Line 431) as shown in the following figure. At harvesting, there were about 70-80 plants in each pot. (3) Irrigation scheduling was added to Lines 435-437.

  1. The irrigation was set at certain percentage of field capacity. How was the field capacity of soil determined? Why loam soil only? Fertilizer application level was based on the results of soil testing. What testing?

Response: (1) Field capacity is 0.30 m3 m-3, which was estimated from the water characteristic curve (Lines 414-415). (2) The soil texture in the Qiliying Experimental Station is loam soil, so only loam soil was used in this experiment (Line 412). (3) The testing is “Soil Testing and Formulated Fertilization” (Line 440).

  1. In the Abstract, all fertilizer level for all treatments were mistaken as 0 kg N hm-2.

Response: Thanks. The fertilizer level was revised. (Lines 15-16)

  1. The language was generally fine, but there were occasional grammatical errors. Some minor editing is required by the authors.

Response: The language was polished.

In summary, I recommended major revision. 

Round 2

Reviewer 2 Report

Comments and Suggestions for Authors

Review of revised version of “Effects of post-anthesis irrigation on the activity of starch synthesis-related enzymes and wheat grain quality under different nitrogen conditions”

The authors have satisfactorily responded to my comments/suggestions and revised the manuscript accordingly. I recommend publishing the revised version.

Comments on the Quality of English Language

The quality of English is generally fine. But The authors should carefully go through the manuscript one more time to polish it before publication.